# mTOR Inhibition via Low-Dose, Pulsed Rapamycin with Intraovarian Condensed Platelet Cytokines: An Individualized Protocol to Recover Diminished Reserve?

**DOI:** 10.3390/jpm13071147

**Published:** 2023-07-17

**Authors:** E. Scott Sills, Conor Harrity, Samuel H. Wood, Seang Lin Tan

**Affiliations:** 1Plasma Research Section, Regenerative Biology Group/CAG, San Clemente, CA 92673, USA; 2Department of Obstetrics & Gynecology, Palomar Medical Center, Escondido, CA 92029, USA; 3Department of Obstetrics & Gynaecology, Royal College of Surgeons in Ireland, D02 HC66 Dublin, Ireland; 4Gen 5 Fertility Center, San Diego, CA 92121, USA; 5OriginElle Fertility Clinic, Montreal, QC H4A 3J3, Canada; 6Department of Obstetrics & Gynecology, McGill University Health Centre, Montreal, QC H4A 3J1, Canada

**Keywords:** mTOR, rapamycin, PRP, platelet cytokines, ovarian reserve, IVF

## Abstract

No major breakthroughs have entered mainstream clinical fertility practice since egg donation and intracytoplasmic sperm injection decades ago, and oocyte deficits secondary to advanced age continue as the main manifestation of diminished ovarian reserve. In the meantime, several unproven IVF ‘accessories’ have emerged including so-called ovarian rejuvenation which entails placing fresh autologous platelet-rich plasma (PRP) directly into ovarian tissue. Among cellular responses attributed to this intervention are reduced oxidative stress, slowed apoptosis and improved metabolism. Besides having an impact on the existing follicle pool, platelet growth factors might also facilitate de novo oocyte recruitment by specified gene upregulation targeting uncommitted ovarian stem cells. Given that disordered activity at the mechanistic target of rapamycin (mTOR) has been shown to exacerbate or accelerate ovarian aging, PRP-discharged plasma cytokines combined with mTOR suppression by pulsed/cyclic rapamycin represents a novel fusion technique to enhance ovarian function. While beneficial effects have already been observed experimentally in oocytes and embryos with mTOR inhibition alone, this proposal is the first to discuss intraovarian platelet cytokines followed by low-dose, phased rapamycin. For refractory cases, this investigational, tailored approach could amplify or sustain ovarian capacity sufficient to permit retrieval of competent oocytes via distinct but complementary pathways—thus reducing dependency on oocyte donation.

## 1. Introduction

Ovarian reserve constrains human fertility potential as female reproductive capacity deteriorates with increasing age. The primordial follicle pool and its rate of activation are compromised by chronic low-grade inflammation over time, culminating in eventual loss of both egg quality and quantity. Once considered nonrenewable, primordial follicles supply the cells which move gradually into the active (growing) follicle group [1,2]. Auditing this process has practical relevance, since few fertility clinics will embark on an IVF cycle without estimating ovarian reserve first. The failure to respond to gonadotropins, irrespective of dose or duration, becomes the final common pathway for any ‘low reserve’ problem.

While choice of IVF stimulation protocols can sometimes be empiric [3], a better understanding of how phosphatidylinositol 3-kinase (PI3K)/mammalian-mechanistic target of rapamycin (mTOR) can be driven by specific gonadotropins has been helpful. Some IVF medications, for example, preferentially boost insulin-like growth factor 1 (IGF-1) with a view to improving oocyte quality [4,5]. Given that IGF-1, epidermal growth factor (EGF), platelet derived growth factor (PDGF) and vascular endothelial growth factor (VEGF) are among the cytokines sourced from activated platelets (PLTs), this led to a hypothesis that reproductive outcome might be shaped by intraovarian platelet-rich plasma (PRP) [6,7,8]. Insulin/IGF-1 signaling can also regulate the PI3K/mTOR cascade to advance follicle stimulating hormone (FSH)-mediated granulosa cell development [9]. In mammalian ovarian function, the PI3K/mTOR pathway interacts with other signaling motifs to calibrate steroidogenesis, granulosa proliferation, corpus luteum survival and oocyte maturation [10,11]. However, satisfactory IVF stimulations (with or without intraovarian PRP) can still, unfortunately, yield cycle cancellation or failure, so an alternative therapeutic option—rapamycin—is explored here for selected, refractory, poor-prognosis IVF patients. While rapamycin is known for its general anti-aging effects [12,13,14], what features would be most salient if this were to be repurposed for an ovarian application, especially if used in tandem with intraovarian PRP?

## 2. PRP and Its Cytokine Constituents

Aging in the ovary unfortunately traces a different trajectory compared to somatic cells, with anti- and pro-longevity genes in oocytes tending to change in opposite directions over time [15]. Ovarian reserve begins to decline measurably often by about 35 years of age and despite advancements of oocyte donation and ICSI [16,17], the need to enlarge the oocyte reservoir remains acute. In this regard, PRP includes components able to upregulate pluripotency genes (e.g., c-Myc, Klf4, Oct3/4, Sox2) associated with reprogramming somatic cells for a pluripotent lineage [18]. Here, Oct4 has special relevance for ovarian remodeling, as human testicular cells express this marker after PRP culture [19], and a parallel response in ovarian tissue is plausible following intraovarian activated platelet-derived cytokines.

Perhaps most crucially, stem cells near the ovarian surface epithelium [20,21] are well placed for research and clinical access. These cells display features which permit cellular transformations into different functional lineages (e.g., epithelium to mesenchyme) [22,23,24]. Local macrophages assert some role in this, although how this influence occurs is not known exactly. Xiao et al. recently (2022) reported that inflammation associated with ovulation in mice drives selective activation of primordial follicles at each estrous cycle [25], depending on follicular macrophages having either M1 or M2 polarization. Interestingly, newborn ovaries cocultured with these macrophage subtypes evince stimulatory features with M1 macrophages but dormancy characteristics with M2 macrophages [25].

This discovery aligned with earlier data which found M2 macrophages more often in older murine retinal tissue [26]. Importantly, this M1/M2 switching is controlled by the PI3K/mTOR signaling pathway [25]. PI3K/mTOR also orchestrates complex intracellular signaling systems, which direct proliferation, cellular quiescence and longevity. With relevance to ovarian rejuvenation practice, PI3K/mTOR is enhanced or regulated by specific PRP cytokine components including EGF, fibroblast growth factor 2 (FGF-2) and IGF-1 [27,28,29]. Such research is consistent with metabolic crosstalk among platelet-derived cytokines and pluripotency networks, perhaps explaining more fully what has become known as ‘ovarian rejuvenation’ [30].

## 3. Rapamycin, mTOR and Reproductive Biology

First described in 1972, rapamycin was isolated from *Streptomyces hygroscopicus* found in soil and plant samples collected on Rapa Nui (Easter Island). First developed as an antifungal and immunosuppressant (see Figure 1), the substance was later found to have potent anti-tumor properties [31]. Further work on its mechanism of action showed that rapamycin complexes with the 12 kDa peptidyl-prolyl cis-trans isomerase FK506-binding protein-12 (FKBP12) to block proliferation and cell growth [32].

In 1991, the protein target of rapamycin (TOR) was discovered in *Saccharomyces cerevisiae*, where TOR gene mutations were noted to cause rapamycin resistance [33]. Subsequent research confirmed mTOR as the allosteric binding site for the rapamycin-FKBP12 complex in mammalian cells [34] where its organizing role in autophagy and cellular senescence was later characterized [35]. It is now agreed that mTOR is a serine/threonine protein kinase in the Class IV PI3K superfamily, which regulates proliferation, growth and cell survival [36].

Integral to many complex signaling networks, mTOR drives adult stem cell proliferation and dictates the differentiation programs of stem cells [37]. Full deletion of mTOR is lethal shortly after embryo implantation [38], yet mTOR null blastocysts can have near-normal early morphology. Nevertheless, trophoblast formation is impaired such that cells taken from the inner cell mass will not proliferate when cultured in vitro. Hence, proper mTOR activity is mandatory for normal embryo development past the blastocyst stage [39]. Rapamycin slows proliferative decay via p16 and butyrate-induced p21 [12], and partial mTOR inhibition enhances maturation of selected populations of human stem cell-derived cardiomyocytes [40].

The consequences of mTOR overactivation are evident in animal progeria models. For example, in the Ercc1-/Δ accelerated aging (mouse) model, rapamycin improved muscle-derived stem cell function via autophagy [41]. Interrupting mTOR signaling by blocking its downstream target (S6K) resulted in longer lifespan and preserved cell function [42]. Specifically, rapamycin was able to recover differentiation and proliferation, reduce senescence and enhance autophagy in a murine progeria model [43]. Hyperactive mTOR with aging thus seems to have serious and harmful consequences for somatic stem cells [13], and mice given rapamycin for the first 45 d of life attain longer lifespans [14]. In human umbilical vein endothelial cells, rapamycin suppresses migration, reverses TGF-β1 stimulated endothelial-to-mesenchymal transitions and downregulates the mesenchymal marker SMA-α [44].

In clinical practice, one profound derangement of aging is confronted in Hutchinson–Gilford syndrome, an ultrarare progeria [45]. In this condition, progerin deposits occur due to a single point mutation c.1824C→T in exon 11 of the LMNA gene [46]. Abnormal primary transcript splicing while forming the lamin A mRNA generates progerin as intracellular accretion; children with this mutation experience accelerated aging with death often before 15 years of age. In this disease, one promising therapy is to stimulate autophagy for clearance of toxic progerin by rapamycin [46]. Of note, rapamycin was recently used for successful treatment of an unrelated cardiomyopathy which involved an LMNA gene variant and dysregulated mTOR [47].

Altered mitochondrial status is another hallmark of aging, and key quality control checks have evolved to prevent vertical transmission of any ovarian mtDNA error. Palozzi and Hurd (2023) recently completed an RNAi screen in Drosophila to find mtDNA integrity surveilled via mTOR complex 1 (mTORC1), implicating the mitophagy adaptor BNIP3 and RNA-binding protein Atx2 as major elements [48]. This extended earlier research that showed that BNIP3 mediates inhibition of mTOR in response to hypoxia [49]. Specifically, Atx2 (*C. elegans* homolog of human ATXN2L and ATXN2) regulates mTOR and the ‘dietary restriction’ phenotype [50]. Indeed, local nutrient availability (e.g., folate [51]) is sensed by mTOR which then coordinates metabolism, growth and autophagy functions. More recently in early mammalian embryos, the impact of rapamycin on mitochondrial fission and mitophagy [52] was studied under varied rapamycin concentrations, with significant improvements noted in blastocyst development, autophagy formation and mitochondrial activation with rapamycin compared to no treatment [52]. In addition, among >200 immature human oocytes submitted for in vitro maturation then fertilized by ICSI, more high-quality embryos were obtained with rapamycin culture vs. untreated controls [53], and histone γH2AX levels (indicating double-strand DNA breaks) in oocytes cultured with rapamycin were also markedly reduced vs. controls [53]. While one-way (permanent) loss of growth potential was previously observed as blocked in arrested cells, rapamycin does not push the arrested cells into proliferation. Instead, rapamycin enables a permissively reversible aging condition [12].

Notwithstanding its use in organ transplantation, which succeeds best with minimal inflammation, rapamycin also paradoxically increases some pro-inflammatory cytokine outputs (i.e., IL-6, IL-12, IL-23) while lowering production of the anti-inflammatory IL-10 [54]. Such actions would not be welcome if rapamycin were planned for simultaneous use with conventional intraovarian PRP, which by design places PLT boluses beneath the surface epithelium [55]. That PLTs exposed to rapamycin undergo functional change is uncontested—a property exploited in vascular stents coated with rapamycin to prevent restenosis [56]. Higher-dose rapamycin interferes with aspirin’s ability to block PLT aggregation [57,58] and more research is needed to know its impact on PLT morphology, membrane phosphatidylserine and thrombin formation [36]. A biphasic disruption in PLT calcium homeostasis does occur with rapamycin, mediated by slowed activation and granule release [59]. Thus, if PLT function were disturbed by rapamycin, then pairing it with conventional PRP would make an odd therapeutic combination.

## 4. PLT Cytokine Augmentation by Rapamycin?

The puzzle of a treatment plan interlocking ovarian PRP with rapamycin, given the latter’s interference with PLT features, may at first seem intractable. However, using condensed PLT cytokines isolated as a cell-free product would bypass the rapamycin tampering problem, and this PRP variant has already been used successfully—without rapamycin—for human ovaries [60]. In other words, if PLT cytokines are separated first and then inserted into ovarian tissue as a filtered condensate by PLT subtraction, the microclimate affected by rapamycin would, in aggregate, end up supporting the potential cytokine commitment of ovarian stem precursors to an oocyte lineage. In this way, rapamycin sustains or possibly extends the ovarian response as entrained by fresh, autologous PLT cytokines.

Investigating how lowering mTOR action affects ovarian biology is not original here, as measurements by NYU researchers found that a two- to four-fold dampening of mTOR activity preserves ovarian function and parity [61]. The current model assumes that limited local inflammation is not injurious but essential, and, from this perspective, intraovarian PLT growth factors followed by rapamycin would be an adaptation of the latter’s recognized role in tissue repair. If introduced at a low dose following ovarian treatment with condensed PLT cytokines, pulsed rapamycin might provide metabolic gains similar to fasting [62] for a nascent follicle pool undergoing induction by platelet-derived growth factors.

## 5. Rapamycin—Scheduling and Toxicity Issues

Rapamycin at relatively low dose is more likely to confer a beneficial response with preferential mTORC1 inhibition, while blunting any undesirable effects on mTORC2 [63]. In the USA, Rapamycin (Sirolimus) is available as 0.5, 1 and 2 mg tablets as well as 1 mg/mL oral solution; it is classified under FDA Pregnancy category C. An improved immune response was reported from one short-term clinical study using the ‘rapalog’ everolimus [64], but a rapamycin protocol specific to the adult human ovary has not been standardized. Unlike high-dose rapamycin used for immunosuppression in transplant medicine, which has received close monitoring, establishing a toxicity threshold for rapamycin use in ovarian (fertility) applications awaits additional study. Poison control records include a failed suicide attempt where a female age 18 consumed more than 100 tablets of 1 mg rapamycin each, and high serum cholesterol was the only documented abnormality [65]. In rats, the LD50 for rapamycin could not be calculated because it exceeded 2500 mg/kg [66].

Inconsistency in rapamycin dosing required for adequate mTOR suppression has also complicated its therapeutic use. Differing rapamycin sensitivities are likely due to varied functional characteristics in mTOR complex 1 and 2 (mTORC1 and mTORC2). For example, mTORC1 is inhibited at low nM levels of rapamycin while mTORC2 suppression usually requires chronic use at higher concentrations [67]. PI3 K/Akt works with Hippo signaling to accelerate recruitment of primordial follicles, while mTORC1 suppression suppresses uncoordinated ‘flash’ discharges resulting in mass follicular activation [9,68]. Platelet-rich plasma (or its condensed cytokine derivatives) plus rapamycin has not been previously studied, probably because both treatments are somewhat novel and available research on each is lacking.

The combined method favored here for ovarian use is a variation on a prior rapamycin dosing calendar with short, pulsed exposure at low doses [64,69]. Specifically, this means phased/cyclic monthly rapamycin with the first day’s oral dose at 3 mg, taken one week after the office PRP procedure. For subsequent days, 1 mg/d is taken for six days and then no rapamycin for the next three weeks. Next, a 3 mg loading dose is repeated on the first day of Cycle #2 again with 1 mg/d taken × 6 d. Cycles #3 and #4 follow the same pattern, so a total of four pulses is completed before planned IVF (see Figure 2).

By comparison, oral rapamycin tablets given for prevention of organ transplant rejection can entail a larger initial first-day 6 mg loading dose followed by 2 mg/d [70]. Baseline and follow-up laboratory measurements appropriate for rapamycin use include serum AMH, complete blood count (w/PLT), C-reactive protein, comprehensive metabolic panel, serum E2, ferritin and FSH/LH, with a fasting insulin and lipid panel. Additional testing for specific gene regulators may not be available in all centers.

## 6. Conclusions

Sponsorship for rapamycin investigation significantly outpaces ovarian PRP research. At present >1000 rapamycin anti-aging clinical trials are formally registered [71], although none focus specifically on premature ovarian insufficiency or low reserve. Given that the total number of registered ovarian PRP clinical trials remains less than ten [72], it is unsurprising that research joining both topics is lacking. Nearly a decade ago, the failure to define general dosing guidelines for rapamycin was acknowledged as a major impediment to its use [73], and consensus on rapamycin use in ovarian biology is likewise absent.

While pulsed or phased oral rapamycin dosing has been discussed in the setting of wound healing [74], the concept described here has not yet been applied in clinical IVF practice. Oocyte capacity sets downstream reproductive fidelity through meiosis, fertilization, nidation and eventual development to term, and other treatments to affect these have been reviewed for the adult human ovary [75,76]. The experimental nature of intraovarian PRP [7] and its condensed plasma cytokines [60] notwithstanding, these interventions alone may still be insufficient for some patients. Given that intraovarian injection of PLT growth factors can increase serum AMH (indicating expansion of the follicle/oocyte unit) [7,77] and mTOR inhibition has been suggested to boost ovarian reserve [78], a bespoke protocol incorporating both might provide a useful synergy.

## Figures and Tables

**Figure 1 jpm-13-01147-f001:**
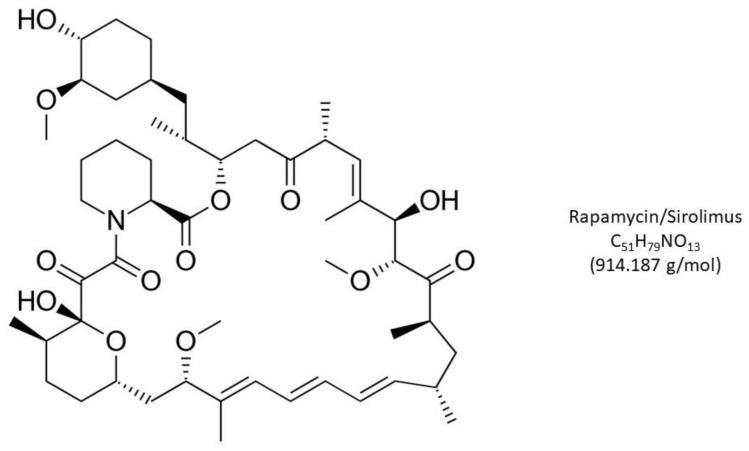
Rapamycin (Sirolimus), a macrolide lactone with potent immunosuppressant, antiproliferative and antifungal properties, received U.S. FDA approval in 1999. While widely used in organ transplant surgery, lower-dose applications of this mTOR inhibitor have successfully decelerated cellular aging to extend lifespan. Any emergent oocyte precursors available after intraovarian PLT cytokine injection may benefit from reduced mTOR activity, as described here.

**Figure 2 jpm-13-01147-f002:**
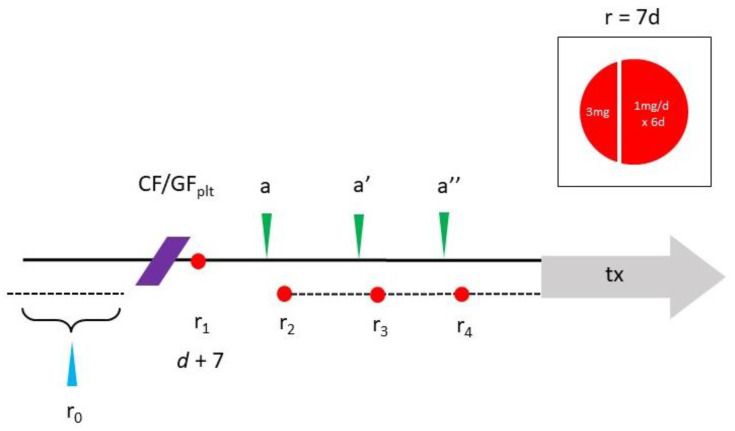
Schematic for combined intraovarian condensed PLT growth factors (CF/GF) and oral cyclic rapamycin (r ×4). Following IRB approval, the study protocol includes enrollment labs (blue arrow) reviewed within 1 mo of ovarian injection (purple). Weekly oral rapamycin phased one-week-per-month begins 7 d after ovarian injection, where a 3 mg loading dose is followed 1 g/d for the next 6 d (inset). Subsequent testing scheduled monthly (a–a″) allows for close monitoring, intended to improve ovarian reserve sufficient for fertility treatment later (tx).

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
