# Peer review of "mTOR Inhibition via Low-Dose, Pulsed Rapamycin with Intraovarian Condensed Platelet Cytokines: An Individualized Protocol to Recover Diminished Reserve?"

_jpm, 2023, doi:10.3390/jpm13071147_

Round 1
Reviewer 1 Report
The article entitled: “mTOR inhibition via low-dose, pulsed rapamycin with intraovarian condensed platelet cytokines: An individualized protocol to recover diminished reserve” is very an interesting and well-written manuscript focussing on important clinical issue i.e. diminished ovarian procreative function.
Based on literature review, expert knowledge and as I presume, their own clinical experience the Authors developed, well documented, justified and described therapeutic protocol for the therapy female infertility associated with low ovarian reserve it is not however clearly stated id the Authors have used this protocol in their every-day clinical practice if so it would be interesting to present what preliminary results.
Author Response
The Referee correctly notes that, as originally written, our paper did not clearly indicate the theoretical nature of mTOR suppression in an IVF stimulation context. We apologize for this oversight, and several changes in response to this deficiency. First, the abstract now includes the word ‘proposal’ (line 26) to emphasize the investigational tone later in that sentence (this conditional theme also appears at lines 58 & 255). In addition, the Conclusion has been modified (line 247) to state directly that this protocol has not yet been used clinically. Perhaps most crucially, the work is now re-categorized not as ‘Article’ but rather as ‘Medical Hypothesis’. Unfortunately, there was a misunderstanding on our part about manuscript classifications beyond the binary Review/Article choice, so this change describes content much better and helps sets readers expectations more properly right from the start.
Reviewer 2 Report
The authors review a novel protocol for increasing ovarian reserve in patients with low-reserve in preparation for IVF. The authors describe the use of PRP administered in the ovarian parenchyma followed by 4 cycles if pulse rapamycin therapy for increasing ovarian reserve. The text is well written and easy to read, with a through and comprehensive review of current knowledge regarding rapamycin use and pathway-activation/inhibition of follicle formation, with a particular focus on mTOR regulation. However, the manuscript is missing a methods section, both in the main text and in the abstract. It is not clear that this is a review.
Author Response
We agree that as a traditional Review or Article, a methods section could not be omitted. However, as noted above, reclassifying the manuscript as a Medical Hypothesis resolves the problem. Another revision concerns the title itself, now formed as a question as way to invite others also to test this protocol and report findings.